# IoT for Monitoring Fungal Growth and *Ochratoxin A* Development in Grapes Solar Drying in Tunnel and in Open Air

**DOI:** 10.3390/toxins15100613

**Published:** 2023-10-15

**Authors:** Charalampos Templalexis, Paola Giorni, Diamanto Lentzou, Francesco Mozzoni, Paola Battilani, Dimitrios I. Tsitsigiannis, Georgios Xanthopoulos

**Affiliations:** 1Department of Natural Resources Management and Agricultural Engineering, Agricultural University of Athens, 75 Iera Odos Str., 11855 Athens, Greece; chartempl@aua.gr (C.T.); dlen@aua.gr (D.L.); 2Department of Sustainable Crop Production (DI.PRO.VE.S.), Università Cattolica del Sacro Cuore, Via Emilia Parmense 84, 29122 Piacenza, Italy; paola.giorni@unicatt.it (P.G.); francesco.mozzoni@studenti.unipr.it (F.M.); paola.battilani@unicatt.it (P.B.); 3Department of Crop Science, Agricultural University of Athens, 75 Iera Odos Str., 11855 Athens, Greece; dimtsi@aua.gr

**Keywords:** *Aspergillus carbonarious*, *Ochratoxin A*, open-air drying, tunnel drying, water diffusivity, grapes, water surface resistance

## Abstract

Optimisation of solar drying to reduce fungal growth and *Ochratoxin A* (OTA) contamination is a crucial concern in raisin and currant production. Stochastic and deterministic analysis has been utilized to investigate environmental indicators and drying characteristics. The analysis was performed using two seedless grape varieties (*Crimson*—red and *Thompson*—white) that were artificially inoculated with *Aspergillus carbonarius* during open-air and tunnel drying. Air temperature (T) and relative humidity (RH) were measured and analysed during the drying experiment, along with grape surface temperature (T_s_), and water activity (a_w_). The grape moisture content, fungal colonization, and OTA contamination were estimated, along with the water diffusivity (D_eff_) and peel resistance (r_peel_) to water transfer. Monitoring the surface temperature of grapes is essential in the early detection of fungal growth and OTA contamination. As surface temperature should be carried out continuously, remote sensing protocols, such as infrared sensors, provide the most efficient means to achieve this. Furthermore, data collection and analysis could be conducted through the Internet of Things (IoT), thereby enabling effortless accessibility. The average T_s_ of the grapes was 6.5% higher in the tunnel than in the open-air drying. The difference between the RH of air and that in the plastic crates was 16.26–17.22%. In terms of CFU/mL, comparison between white and red grapes in the 2020 and 2021 experiments showed that the red grapes exhibited significantly higher values than the white grapes. Specifically, the values for red grapes were 4.3 in 2021 to 3.4 times in 2020 higher compared to the white grapes. On the basis of the conducted analysis, it was concluded that tunnel drying provided some advantages over open-air drying, provided that hygienic and managerial requirements are met.

## 1. Introduction

Grapes are one of the most widely grown fruits, with an annual growth rate of almost 75,000 tonnes. Approximately 50% of grapes are utilized in the wine industry: one-third is consumed as fresh, and the remainder is dried. Due to their high moisture content, they are susceptible to spoilage after harvest, even during cold storage. High sugar content and high moisture content stimulate the growth of microorganisms, leading to their spoilage. Drying is an effective method to decrease moisture content to an acceptable level and reduce the risk of microorganism growth. Furthermore, low moisture content decreases the potential for enzymatic degradation and moisture-mediated harmful reactions, making drying an optimal approach for grape preservation. Currently, in various developing countries, the commonly employed grape drying methods are open-air solar drying and drying under shade [1].

Grapes are prone to diseases caused by non-obligate fungi such as *Aspergillus* section *Nigri*, *Penicillium* spp., and others. Fungal invasion depends on grape maturity and quality state. *Alternaria*, *Cladosporium*, *Botrytis,* and *Rhizopus* are fungi detected at early veraison, whereas *Aspergillus* and *Penicillium* are at harvest and during sun drying. *Aspergillus carbonarius* and *A. niger* (black *Aspergilli*) are known to be responsible for *Ochratoxin A* (OTA) contamination in grapes and raisins [2]. Factors significantly affecting fungal growth during harvesting and storage are temperature, water activity (a_w_), and period that processed products remain under conditions favourable for fungi proliferation. Other factors conducive for fungal development are the presence of fungal spores, mechanical damages, presence of pest insects, storm and rain damages, moisture stress, mineral deficiency, pH, O_2_ and CO_2_ levels, chemical and physical treatments, and the product drying and re-wetting speed for some commodities. Mechanisms affecting fungi proliferation and OTA production are not the same. Therefore, OTA can be detected even in the absence of visible signs of fungal presence. The grape growing region can greatly influence OTA contamination, exhibiting an increasing trend moving from west to east and from north to south in Europe [3]. *A. carbonarius*, as the main OTA producer, takes the lead over *A. niger* during drying, as it is more adaptable to decreasing a_w_ [4]. The ecological parameters of black *Aspergilli* (*A. niger* and *A carbonarius*) have been extensively studied. The acquired information is crucial for the development of risk models amidst dynamic and interdependent environmental parameters.

*A. carbonarius* has been reported as the main source of OTA contamination in wine and dried vine fruit. Commission Regulation (EC) No 915/2023 [5] has set a maximum OTA content of 8 μg/kg in currants, raisins, and sultanas. *A. carbonarius* and *A. niger* produce unicellular conidia with *melanin* and *aspergilline* in their cell walls but differ in their UVC resistance and occurrence in grapes [6]. The higher UVC resistance of *A. carbonarius* spores, compared to those of *A. niger*, explains the high incidence of *A. carbonarius* on grapes during prolong sun exposure [7]. Infrared imaging is a promising technique for the early detection of postharvest diseases in fruits. By monitoring the thermal radiation emitted by the fruit, infrared imaging enables the identification of subtle temperature variations associated with disease development. Infrared imaging has emerged as a promising tool for the detection of *A. carbonarius* during the solar drying of grapes. This fungal pathogen is known to produce mycotoxins that can contaminate dried fruits. By capturing the thermal radiation emitted by the drying grapes, infrared imaging allows the identification of temperature variations associated with the presence of *A. carbonarius*. The reading of infrared sensor data through an internet address has revolutionized the way we interact with and monitor our surroundings. With the advancement of Internet of Things (IoT) technology, infrared sensors can now be seamlessly integrated into networked systems, allowing for remote access and real-time monitoring of data. Infrared sensors capture and measure infrared radiation, providing valuable information about temperature, motion, and other environmental factors. By connecting these sensors to an internet address, users can remotely retrieve and analyse the data collected by the sensors from anywhere in the world. This capability has numerous applications across various industries, including smart home automation, industrial monitoring, and environmental sensing. It enables efficient control, predictive maintenance, and timely decision-making based on the transmitted sensor data. Overall, the integration of infrared sensors with internet addresses enables a new level of convenience, accessibility, and efficiency in data monitoring and management [8].

This study analyses important key aspects of solar drying of grapes in open-air and in tunnels by employing IoT infrared imaging to measure grape surface temperature. Pros and cons were identified, and were analysed, proposing improvement measures. Analysing critical factors, including air temperature, humidity surrounding the grape drying bed, a_w_, drying time, and grape drying properties (water diffusivity and peel resistance to water transport), as well as colony forming units (CFU) and OTA production, provided essential information to understand the underlaying mechanisms during open-air and tunnel drying of grapes.

## 2. Results and Discussion

### 2.1. Stochastic Analysis of Drying Data Collected during the Drying Period

The data collected during the drying period were analysed throughout the 24 h monitoring period, and the average, minimum, and maximum values are tabulated in Table 1. The average temperatures of the grapes (control and infected) in the tunnel and in open air were in a narrow range, from 22.40 to 25.72 °C, with limited significant differences (*p* ≤ 0.05). The grapes in the tunnel had higher maximum surface temperatures than those dried in open air, with a wide range from 6.2 to 14.4 °C. Similar temperature differences were found in the air temperatures monitored in the plastic crates (Table 1), ranging from 19.86 °C for red grapes to 16.29 °C for white grapes. The relative humidities in the air and inside the plastic crates (red and white grapes) were statistically different (*p* ≤ 0.05), showing a difference between 16.26% and 17.22% (Table 1); therefore, the choice of relative humidity monitoring points to be used in predictive models for *A. carbonarius* proliferation and OTA production should be spatially focused. There were no significant differences (*p* ≤ 0.05) in relative humidity between red and white grapes in the crates per drying case (Table 1).

Grape temperature is an important factor in fungal infection along with a_w_ because, in the absence of surface wounds, infection spreads over the surface of the berry [9]. In the 2020 experiment, surface measurements were taken three times a day (9.30–14.30–18.30). In the 2021 experiment, grape surface temperature was monitored using the infrared sensors, providing hourly data. Table 2 shows the average and standard error of the 24 h grape surface temperatures with respect to the difference between the control and infected samples. Temperature differences exceeding 1.0 °C between the control and infected grapes were observed during daylight hours from 09:00 to 20:00 when the average total solar radiation exceeded 300 W/m^2^. Deviations from this response were also observed in some cases, and the reasons for these deviations will be elaborated on.

Environmental conditions (air temperature, rainfall, and relative humidity) are important for black *Aspergilli* growth and OTA production. *A. carbonarius* grows optimally at 30–35 °C, and a_w_ = 0.92–0.98 [10]. For OTA production, optimum conditions are a_w_ = 0.95–0.98 and air temperatures of 15–20 °C or 30–35 °C, depending on the strain, but independent of geographical origin [10].

In this experiment, the infrared sensors were placed outdoors to monitor the surface temperatures of the drying grapes (Figure 1). By focusing the analysis on the daylight period (09:00–18:00) when significant differences in total solar radiation (>300 W/m^2^) and average grape surface temperature were recorded between the control and infected grapes (Table 2), the average, upper (UCL), and lower (LCL) confidence limits per case studied were estimated and presented; open-air drying/white grapes: 2.206 °C, 3.554 °C, and 0.857 °C; open-air drying/red grapes: 2.145 °C, 3.518 °C, and 0.772 °C; open-air drying/white grapes: 2.206 °C, 3.554 °C, and 0.857 °C; tunnel drying/white grapes: −0.865 °C, 1.644 °C, and −3.375 °C; and tunnel drying/red grapes: 2.308 °C, 3.775 °C, and 0.842 °C. Although statistical analysis (t-test) did not show a significant difference between the average values of total solar irradiance in 2020 and 2021, the values for 2020 were up to 33.4% higher than those for 2021.

Regarding the previous analysis, it can be seen that the average value of the surface temperature difference between control and infected grapes was above 2.1 °C in all cases, except for the tunnel drying/white grape case. The tunnel drying in the white grape case had a significantly lower value. The accuracy of the infrared sensors was ±1.5 °C, which justified the significance of higher than 1.5 °C estimated averages. The average CFU/mL values per tested case related to the difference in grape surface temperatures (T_s_|_control_ − T_s_|_infected_) is shown in Figure 2. This correlation showed that there is a limit to the CFU/mL below, of which the resulting difference in grape surface temperature (T_s_|_control_ − T_s_|_infected_) is equal to or less than the accuracy of the infrared sensors and therefore insignificant. This is important for the validity of the method; the CFU/mL threshold corresponding to the accuracy of the infrared sensors in measuring the average surface temperature difference (1.5 °C) is 545.2. From this analysis, it is clear that the average CFU/mL was below this limit (545.2) and was actually 166.5 in case 3 (tunnel drying/white grapes). This point is very important for the operation and accuracy of the infrared monitoring of grapes’ surface temperatures and deserves further analysis. Monitoring grape surface temperature using infrared sensors can introduce a bias in temperature surface measurements due to uncontrolled environmental conditions, such as air drafts or water condensation at night. To address this issue, the IACT [11] protocol was employed for infrared sensing in medical cases, in the absence of a dedicated protocol applicable to infrared sensing during grape drying. In this context, guidelines are presented to enhance measurement accuracy and repeatability. To achieve a steady state infrared measurement, it is necessary that the measurement area is draft free, and the surrounding air temperature varies gradually. It is also recommended that the air humidity around the area is such that it avoids condensation on the surface that could influence radiant infrared energy. IACT [11] also suggests that infrared sensors must meet a minimum standard of precision and repeatability of 0.1 °C and an accuracy of ±2% or less for detecting temperature differences for reliable and reproducible results. The latter specification applies to high-end instrumentation used to detect small temperature differences on the surface during the early stages of fungal growth.

Similar to the surface temperatures of the drying grapes, it is very important to know the a_w_ of the grapes. For this purpose, the well-known G.A.B. sorption model was adopted. The M_o_, C, and K_b_ were estimated by the *Levenberg–Marquardt* optimisation algorithm (Table 3), where M_o_ was the monolayer moisture content; C was a constant related to the heat of sorption of the first layer; and K_b_ was related to the heat of adsorption of the multi-layer.

Statistical analysis resulted in $R_{\mathrm{adj}}^{2}=$ 99.95% and SEE = 0.053 (*p* ≤ 0.05). The experimental (points) and predicted (lines) a_w_ values are shown in Figure 3. In addition to the standard fit criteria ($R_{\mathrm{adj}}^{2}$, SEE), positive parameters (M_o_, C, and K_b_) were found, and the values were significant, as the asymptotic lower and upper 95% confidence limits did not include zero. The deviation of a_w_ between experiments 2020 and 2021 was 4.95 for the a_w_ range 0.00–0.99 and 6.19 for the a_w_ range 0.95–0.98; the deviation was evaluated in terms of root mean square error (RMSE).

At the time of sampling on 14 September 2021 (146 h after the start of the drying experiments), the estimated a_w_ was lower than 0.6 for red and white grapes dried in the tunnel, but not for the corresponding grapes dried in the open air, where the drying time was longer. In particular, white grapes dried in the open air had an a_w_ close to 0.80, whereas red grapes had an a_w_ close to 0.70. Nevertheless, the OTA contamination was higher in open air (Table 4) in the 2020 experiments compared to the 2021 experiments. The data tabulated in Table 5 were also interesting, where the CFU/mL values were higher in the tunnel than in the open air in the 2021 experiments, with a trend significantly opposite to that of 2020. Comparing the CFU/mL between white and red grapes, in both experiments (2020 and 2021), the red grapes showed much higher values compared to the white ones, ranging from 4.3 times in 2021 to 3.4 times in 2020.

The mechanisms favouring fungal growth are different from those favouring OTA production/accumulation. The analysis of variance (ANOVA) and multiple comparison tests of two years of solar drying experiments produced results from which conclusions should be drawn with great caution. A different mathematical analysis was tested to analyse this problem and produce more understandable results. The CFU/mL and OTA values estimated from the three samplings and the eight drying cases were fitted to a *Weibull* distribution. This distribution was used to describe the behaviour of biological systems that have some degree of temporal variability due to time-dependent stress conditions. The goodness-of-fit test for the CFU/mL and OTA values showed that the estimated D-values (see in Table 6) were greater than 0.05, and therefore the assumption that the “distribution of CFU/mL and OTA values during drying fall into *Weibull* distribution” could not be rejected with 95% confidence. The shape factor of the *Weibull* distribution determines the shape of the curve (concave or convex): if β < 1, the failure rate decreases over time; if β = 1, it is constant over time; and if β > 1, it increases over time (an “ageing process” takes place). The higher the β value, the faster the process deteriorates. The shape factors for the CFU/mL and OTA (Table 6) showed a significant difference. In particular, the shape factors of CFU/mL ranged from 0.382 to 2.619, whereas the corresponding values for OTA ranged from 2.625 to 10.752. Comparing the shape values of CFU/mL with those of OTA, it can be seen that the shape factor of OTA for white grapes is 5.3 times greater on average than that of CFU/mL, and 7.8 times greater for red grapes. This trend highlights the difference in the mechanisms affecting the rate of development of *A. carbonarius* in relation to OTA accumulation in the grapes. This analysis of CFU/mL and OTA using the *Weibull* instead of the normal distribution is better because the former can also model skewed data. Its flexibility favours the modelling of both left and right skewed data, allowing efficient analysis in a wide range of problems.

### 2.2. Deterministic Analysis of the Drying Process

The first step in modelling the drying process of grapes is to well design the domain in which the deterministic model will be set up and run. The grapes were assumed to be prolate spheroids with two axes (a major and a minor axis) that shrank asymmetrically during the drying process [12]. The shrinkage velocity (*m*/*s*) per drying case was estimated by image analysis (Table 7) and used as input to the model. The drying data of all experimental cases (Table 7) were analysed to estimate their drying properties by computational simulation, as presented by Templalexis et al. [12]. The drying samples used for weighing (20 grapes per drying case) were photographed daily, and the digital photos were processed using Adobe Photoshop, v.13012 (Adobe Photoshop Inc., San Jose, CA, USA.) to evaluate the shrinkage effect (Table 7).

The physical problem under consideration gives rise to a computational model for deriving theoretical predictions of the spatio-temporal distribution of water content in drying whole grapes as a function of effective water diffusivity (D_eff_), peel resistance to water transfer (r_peel_), and shrinkage. The governing equations with boundary and initial conditions were numerically discretised by the Finite Element Method (FEM) using COMSOL Multiphysics 5.1. The unstructured mesh consisted of 1600–1800 free quad elements (Figure 4).

In this study, the D_eff_ and the surface mass transfer coefficient (k_c_) were estimated by solving the *Fick’s* law of diffusion problem using the experimental drying curves, [MC = *f*(t)] [12]. The *Levenberg–Marquardt* optimization algorithm and the FEM were combined to estimate D_eff_ and k_c_ in an inverse mass transfer problem related to the drying of grapes with shrinkage. The time-dependent problem was solved by an implicit time-stepping method, the Backward Differentiation Formula (BDF), whereas the resulting system of nonlinear PDEs (Partial Differential Equations) in the time–space domain was solved by coupling the FEM with the Arbitrary Lagrange–Eulerian (ALE) method to account for shrinkage [12]. The boundary conditions (see Figure 4) controlled the displacement of the moving mesh with respect to the initial geometry. The linearised equation system was solved using the Parallel Sparse Direct Solver (PARDISO), which is faster than other available linear solvers [12]. Inverse problems have multiple possible solutions rather than a unique solution, which makes their solution prone to error if only the minimisation of the objective function is considered. Three additional criteria were used in the optimisation process:-The D_eff_ should be lower than those for self-diffusion of water (3.6 × 10^−9^ m^2^/s (at 45 °C), 4.37 × 10^−9^ m^2^/s (at 55 °C) and 5.09 × 10^−9^ m^2^/s (at 65 °C)) [12].-The D_eff_ should be in the range of 10^−11^–10^−9^ m^2^/s [13].-The mean relative error (MRE) between the estimated and experimental water content should be less than 10%.

The optimisation procedure was based on the input values tabulated in Table 2. The estimated k_c_, D_eff_, and surface resistance to water transport (r_peel_ = 1/k_c_) for the drying cases tested are seen in Table 8.

As can be seen, D_eff_ was 244% on average (54% in 2020) higher in the tunnel than in the open air. The values in the parenthesis are those from the 2020 experiments conducted by Templalexis et al. [12]. This can be partly explained by the temperature, which was higher in the tunnel than in the open-air (Table 1). Comparing D_eff_ between infected and control samples in tunnel drying, infected grapes had more than 50% (70% in 2020) higher D_eff_ than the control grapes. In the case of the open-air drying, the infected grapes had more than 120% (60% in 2020) higher D_eff_ than the control grapes. As can be seen in Table 8, D_eff_ was on average 92% higher in the 2020 experiments compared to the 2021 experiments. In particular, the average temperature was 70% higher in the 2020 experiments compared to the 2021 experiments (open-air drying: 77%; and tunnel drying: 63%). This is also explained by the duration of the drying process, which is discussed below. In the light of the previous discussion, and the data in Table 5, where the accumulated OTA was 16% higher in the open-air drying in the 2020 experiments compared to the 2021 experiments and 48% higher in the tunnel drying, it can be concluded that higher temperatures may favour the drying process but, on the other hand, also favour the production/accumulation of OTA. Although it is difficult to draw simple conclusions in such a complex and multi-parametric problem, there must be an optimum combination of temperatures in both drying processes (open air and tunnel), which, on the one hand, favours the drying process and, on the other hand, causes the least OTA accumulation. The mechanisms of water transport are complex since the drying rate is affected by the biology/anatomy of the grape berry, the resistance of the peel to water transport of each grape variety, and the degree of ripeness. The results from the analysis of 2020 and 2021 experiments are consistent with the fact that *Aspergillus* growth is favoured in red grapes compared to white grapes. The grape peel is covered by a waxy layer (cuticle) and has few functional stomata, so water loss is mainly through the waxy cuticle at a relatively slow rate. When the rate of water loss increases due to high temperatures, splitting of the peel occurs. Although macroscopic observations did not agree this phenomenon in both years (2020 and 2021), it cannot be overlooked in the absence of microscopic observations. When the peel is damaged, nutrients are no longer restricted, and the microbial population increases dramatically [9]. Ramla et al. [14] reported that the effect of dipping pre-treatment on OTA accumulation in sultanas (white grapes) and currants (red grapes) was more severe in red grapes than in white grapes, and this behaviour was mainly due to the components of red dried grapes that may favour fungal growth, leading to faster and higher OTA accumulation. The r_peel_ showed a different response during drying depending on the treatment tested (2021 experiments). In addition, significant differences were found between the 2020 and 2021 experiments. The r_peel_ was ≈1.4 (2.4 in 2020) times lower for grapes dried in the tunnel than for those dried in open-air (Table 8). The difference in r_peel_ between the 2020 and 2021 experiments was 20.84% higher on average in 2020 in the open-air experiments but 32.04% lower in the tunnel drying experiments. The previous differences in r_peel_ are consistent with the differences in estimated CFU/mL between 2020 and 2021 (Table 4), but not in a quantitative way. This response is consistent with the fact that the mechanisms favouring CFU/mL and OTA production/accumulation are different. The r_peel_ response was consistent with the drying time, which was 407 h (390 h in 2020) in the open-air drying and 290 h (220 h in 2020) in the tunnel, considering that drying was stopped when no mass loss was observed. In tunnel drying, the r_peel_ of the control white grapes was 12.8% higher than that of the infected white grapes, and the r_peel_ of the control red grapes was insignificant compared to that of the infected red grapes. In Table 1, the average drying temperature in the tunnel was 24.8 °C, 6.5% higher than in the open air, which was 23.3 °C. The drying temperatures in this solar drying experiment were lower than the temperatures (>50 °C) used in artificial drying. These results were derived from computational simulations and need to be validated by electron microscopic analysis of peel disintegration during the solar drying in order to evaluate the extent of disintegration per drying method, as well as the effect of *Aspergillus* growth and OTA development on grape peel disintegration. In Figure 5 and Figure 6, the drying curves MR = *f*(t) along with the variation in OTA with drying time (t) are presented. The grey shaded area is the optimum zone (MR = 1.0, a_w_ = 0.98; and MR = 0.27, a_w_ = 0.92) for *A. carbonarius* growth. The initial water content of red grapes was higher than that of white grapes in both years (2020 and 2021). The water content of red grapes was 5.89 kg_water_/kg_dm_ (4.55 kg_water_/kg_dm_ in 2020), and that of white grapes was 4.02 kg_water_/kg_dm_ (3.95 kg_water_/kg_dm_ in 2020).

In terms of OTA production, open-air drying showed a non-significantly higher content compared to tunnel drying, although the opposite trend was found in 2020 (Table 5). This could be the result of the similar reduction rate of grapes a_w_ in tunnel and open-air drying. This response did not favour a_w_ × temperature induced stress capable of favouring OTA production, as it happened in the 2020 experiments (Table 5 and Figure 5 and Figure 6).

Red grapes showed non-significantly higher fungal contamination compared to white grapes (*p* ≤ 0.05), even with almost four times higher CFU/mL (Table 5). OTA contamination showed a slight increase with drying time at the final sampling compared to the middle and initial sampling (*p* ≤ 0.05), although the rate of increase was much higher during the 2020 experiments (Table 5). At the end of the experiments, the OTA levels in red grapes dried in the tunnel (4.32 µg/kg) and in the open air (4.34 µg/kg) were higher than those in white grapes dried in the tunnel (2.45 µg/kg) and in the open air (3.54 µg/kg), although the amount of OTA did not exceed the OTA limit of 8 µg/kg for currants, raisins, and sultanas in all cases. The corresponding OTA levels shown in Table 4 for the experiments of 2020 were higher, in some cases more than twice. Based on the statistical analysis regarding OTA, significant interactions were reported between grape variety and type of drying (*p* ≤ 0.05) in both years 2020 and 2021 (Table 4). The type of drying influences the drying rate, due to higher drying temperatures achieved and the consequently shorter drying time, which affects the production of CFU/mL and OTA. Grape variety has also been found to favour OTA accumulation in red grape varieties compared to white varieties, probably due to the components (i.e., Brix and acidity) of red dried grapes [10,14], but also due to the higher water content, as found in both experiments (2020 and 2021). 

## 3. Conclusions

This study utilises data fusion from three distinct yet interconnected scientific fields such as “Phytopathology”, as OTA accumulation is attributed to *A. carbonarius* proliferation, “Software and hardware engineering”, through the adoption of infrared sensing and IoT protocols for continuous monitoring of grape surface temperature and easy access to collected data and “Computational Fluid Dynamics”, as drying properties were estimated from computer simulation based on the fundamental principles of “Transport phenomena”. To achieve reliable and reproducible grape surface temperatures through the use of infrared sensing, it requires adherence to specific guidelines during measurement. To ensure accuracy, the measurement area should be draft-free, and the air humidity should not allow formation of condensation on the measurement surface, which can affect the radiant infrared energy. Also, it is essential to use high-end instrumentation capable of detecting even the smallest temperature differences on the surface during the early stages of fungal growth. This study investigated the correlation between the average CFU/mL values per tested case and the difference in grape surface temperatures (T_s_|_control_ − T_s_|_infected_). From this correlation, it was found that the measured difference of grape surface temperatures (T_s_|_control_ − T_s_|_infected_) becomes equal to or less than the accuracy of the infrared sensors once the CFU/mL values fall below a threshold. It would be informative to extend this investigation using infrared sensors with varying measurement accuracy. This will enable testing of the minimum CFU/mL levels, at which infrared sensors can detect significant differences between infected and healthy grapes during open-air and tunnel drying. This will be highly significant in developing and applying this method for the early detection of *A. carbonarius* and OTA production.

Over the course of two consecutive years (2020 and 2021), this study aimed to identify key measurements necessary for early detection of *A. carbonarius* and OTA production in open-air and tunnel drying. Various properties of the surrounding air, drying grapes, and drying properties through Computational Fluid Dynamics simulation (D_eff_ r_peel_) were measured to achieve this objective. The analysis of all the collected data revealed that the selection of environmental indicators (such as air or grape surface temperatures, relative humidity of air or inside the crates during grape drying, and solar radiation) for feeding as inputs predictive models of OTA production should be approached with caution. This is due to the significant spatiotemporal variability exhibited by the monitored indicators, which could result in significant bias in OTA modelling. Estimating D_eff_ and r_peel_ can help in comprehending the mechanisms involved in growth of *A. carbonarius* and OTA production. These two properties describe the movement of water from the endocarp and mesocarp of grapes to their skin (exocarp). This discussion holds significance, as D_eff_ and r_peel_ explain the water transport in grapes for the different grape varieties, drying methods, and between healthy and infected grapes.

## 4. Materials and Methods

### 4.1. Plant Material

Twin solar drying experiments were carried out on red (var. *Crimson seedless*) and white (var. *Thompson seedless*) grapes in the open air and in a tunnel from 08 September 2021 to 23 September 2021 to investigate the role of grape surface monitoring throughout the drying process as well as the environmental conditions and resulting drying properties in *A. carbonarius* infection and OTA contamination. In order to carry out this study, red and white grapes were taken from a local market and on arrival at the laboratory, defective berries (injured, overripe, etc.) were discarded, and the remaining berries were divided into two batches: healthy (hereafter referred to as “Control”) and the infected (hereafter referred to as “Infected”) grapes; the latter were artificial inoculated according to the following experimental protocol. The grape berries were arranged in a single layer within shallow, perforated plastic crates (see Figure 1) in a sparsely populated manner. This was performed to guarantee uniform drying conditions for all the grapes in each crate, hence making the measured surface temperature of the grapes to be representative of the complete mass.

### 4.2. Preparation of Fungal Inoculumn and Treatment Samples

A strain of *A. carbonarius* (ITEM 5012), previously tested for its ability to produce OTA and kept in the official fungal collection of the Institute of Sciences of Food Production of the National Research Council (ISPA–CNR) in Bari and in the fungal collection of the Department of Sustainable Crop Production (Di.Pro.Ve.S.) of the Università Cattolica del Sacro Cuore in Piacenza, was used for inoculum preparation. The strain was inoculated on Petri dishes containing Potato Dextrose Agar (PDA, Biolife, Milan, Italy) and incubated for 7 days at 25 °C (12 h light photoperiod). At the end of incubation, the dishes were washed with 10 mL of sterile distilled water, and the suspension obtained was adjusted to a concentration of 10^4^ conidia/mL. Each sample group (open-air and tunnel drying, white and red grape bunches) was immersed in the conidial suspension for 5 min and allocated for drying. For comparison, untreated grapes (not inoculated with fungal inoculum) were also included and subjected to solar drying until steady weight was achieved. The grapes were sampled at three sampling times: the beginning of the experiment (7 September 2021), the middle of the experiment (14 September 2021), and the end of the experiment (7 October 2021). At each of the three sampling times, 20 berries were randomly selected in triplicate from the tested clusters, i.e., 3 sampling times (beginning-middle-end) × 2 grape varieties (white-red) × 2 drying treatments (Control–Infected) × 2 drying methods (open air–tunnel drying) × 60 (3 × 20) berries = 1440 berries were analysed in total. After grape sampling, the grapes from different treatments were stored separately in plastic bags at −18 °C until the later analysis of fungal colony forming units (CFU/mL) and quantification of OTA (μg/kg).

### 4.3. Fungal Infection of Grapes and Detection of Ochratoxin A

Grapes from the different sampling groups were crushed to obtain must, consisting of juice and berry residue with peel, which was used for both CFU and OTA analysis. The must (1 mL) was added to 9 mL of 1% peptone water, homogenised by vortexing and the suspension was used for serial dilutions from 10^−1^ to 10^−6^, plated on Potato Dextrose Agar (PDA, Biolife, Milan, Italy) amended with cloramphenicol (0.5%), and incubated at 25 °C for 5 days (12 h light photoperiod); the test was carried out in triplicate. The developed black *Aspergilli* colonies were counted and the results expressed as colony forming units per mL of grape must (CFU/mL). For OTA analysis, the musts were extracted with ethanol (70%) in a 1:1 (*v*:*v*) ratio, and the mycotoxin concentrations were determined by Enzyme-Linked Immunosorbent Assay (ELISA). The sample extracts were analysed using AgraQuant *Ochratoxin A* (RomerLabs, Getzersdorf, Austria) for OTA quantification. Mycotoxin extraction and analysis were carried out according to the manufacturer’s instructions and considered reliable [12].

### 4.4. Hygrothermal Measurements during Drying Experiments

During the solar drying experiments, a hygrothermal sensor and data logger, Hobo U10–003 (Onset Computer Corp., Bourne, MA, USA), was used to monitor the air temperature and relative humidity every 10 min, with a resolution of 0.4 °C and 0.5%, and an accuracy of ±0.7 °C and ±3.0% for temperature and relative humidity, respectively. 

Grape surface temperature was monitored using an array of infrared temperature thermometers Optrics CS LT (Optris GmbH, Berlin, Germany) (Figure 1) with accuracy ± 1.5 °C (operating range −50 °C to +1030 °C), and infrared repeatability ± 0.75 °C. Temperature measurements were taken at 5 min intervals, and data were recorded and stored in a data logger. Grape weighting was carried out twice a day (morning 09.30 and evening 16.30) in the eight experimental cases with 20 grapes per case (160 grapes in total).

The system for collecting and recording the measurements was developed at the Laboratory of Farm Machine Systems of the Agricultural University of Athens, Greece. It consisted of a motherboard responsible for displaying, processing, and storing the measurements and a number of 1 to 8 daughterboards for connecting the sensors. Each daughterboard could read up to 8 analogue sensors with a resolution of 10 bits and a sampling rate of 10 Hz. The daughterboard was based on the ATMEGA32-16PU processor from MICROCHIP (ATMEL). The motherboard was based on ST’s 32F746GDISCOVERY board with STM32F746NGH6 processor and included 1024 KB ROM (Flash), 320 KB RAM, a 4.3-inch touch screen, 4.3-inch SD card, Ethernet, and am RS485 port. The connection between the motherboard and the daughterboard could be wired (RS485) or wireless (Digi XBee 802.15.4 at 2.4 GHz). The software for both boards was written in C and C++. The measurements were recorded on the motherboard’s SD card and in a database (MySQL) on the laboratory’s server. A browser application (WebBrowser) in HTML and JavaScript and an Android mobile application (apk) in Java for Android were developed to monitor the measurements in real time.

The a_w_ was measured using a HygroLab C1 (Rotronic AG, Bassersdorf, Switzerland) equipped with an HC2–AW sensor at 25 °C. The instrument was calibrated in the range of 0.65–0.95 using SCS certified humidity standards EA65–SCS, EA80–SCS and EA95–SCS. The a_w_ was related to the respective moisture content using the statistical programme Statgraphics 19 (Statgraphics Technologies, Inc., The Plains, VA, USA) and the Guggenheim, Anderson, and de Boer (G.A.B.) sorption isotherm model (Equation (1)). The parameters M_o_, C, and K_b_ were estimated using the Levenberg–Marquardt optimisation method,
(1)$$M_{e}=\frac{M_{o}\;C\;\;K_{b}\;a_{w}}{\left(1-k_{b}\;a_{w}\right)\left(1-k_{b}\;a_{w}+C\;\;K_{b}\;a_{w}\right)}$$
where M_o_ is the monolayer moisture content; and C and K are the adsorption constants, related to the interaction energies between the first and the subsequent sorbed molecules at each sorption site. The a_w_ was measured in triplicate on red and white control grapes (tunnel and open air) according to a rotation scheme at the end of each day of the experiment using the “dynamic method”. The average initial moisture content of the grapes was 6.28 ± 0.8 kg_water_/kg_dm_ (red, tunnel drying), 3.85 ± 0.2 kg_water_/kg_dm_ (white, tunnel drying), 6.84 ± 1.0 kg_water_/kg_dm_ (red, open-air drying), and 3.90 ± 0.2 kg_water_/kg_dm_ (white, open-air drying) and was estimated gravimetrically at the end of the experiments at 105 °C for 24 h [15]. Grape drying was considered complete when the moisture ratio (MR) was less than 0.1.

### 4.5. Statistical Analysis and Weibull Distribution

The three-parameter *Weibull* distribution is a well-known continuous probability distribution used to describe the behaviour of systems or events that have some degree of temporal variability. The *Weibull* distribution was developed to analyse the strength of materials and the resulting failure with time under stress conditions but has since been used to describe the kinetics of quality control, biology, enzymatic, and chemical degradation. The probability density function of the *Weibull* distribution is described as follows [16].
(2)$$f\left(x\right)=\{\begin{array}{l} \frac{\beta }{\alpha }{\left(\frac{x-\gamma }{\alpha }\right)}^{\beta -1}\exp\left[-{\left(\frac{x-\gamma }{\alpha }\right)}^{\beta }\right],\;\;X>0\\ 0,\;\;\;\;\;\;\;\;\;\;\;\;\;\;\;\;\;\;\;\;\;\;\;X\leq 0\; \end{array}$$
with α > 0 and β > 0, where α is the scale parameter as a reaction rate constant, β is the shape factor as a behaviour index, and γ is the location parameter, which locates the distribution along the abscissa. Changing the value of γ has the effect of “shifting” the distribution and its associated function either to the right (if γ > 0) or to the left (if γ < 0). The Weibull distribution is reduced to first-order decay/growth kinetics when β = 1. The failure rate for the Weibull model is an increasing function of time for β > 1 (concave downward) and a decreasing function of time for β < 1 (concave upward). When β = 1, the distribution becomes exponential, and the failure rate becomes constant. The shape factor (dimensionless) determines the shape of the curve (concave or convex), whereas the scale parameter is a rate constant. Therefore, if β < 1, the failure rate decreases over time; if β = 1, the failure rate is constant over time (random external events cause degradation); and if β > 1, the failure rate increases over time (an “ageing process” takes place) [12].

### 4.6. Data Analysis

Statistical analysis of experimental data was carried out by Statgraphics 19 (Statgraphics Technologies, Inc., The Plains, Virginia, USA). ANOVA was used to estimate the average values of CFU/mL, OTA (μg/kg) detected in infected grapes (red and whites), and measured temperatures (air and grape surface). Fisher’s Least Significant Difference (LSD) test was used to identify statistically significant differences between averages. This test is liberal in terms of comparison wise error rate or type I error, compared to other multiple comparison tests (Duncan, Bonferroni, Tukey, and Scheffe), but it is powerful in detecting true differences between averages [17].

## Figures and Tables

**Figure 1 toxins-15-00613-f001:**
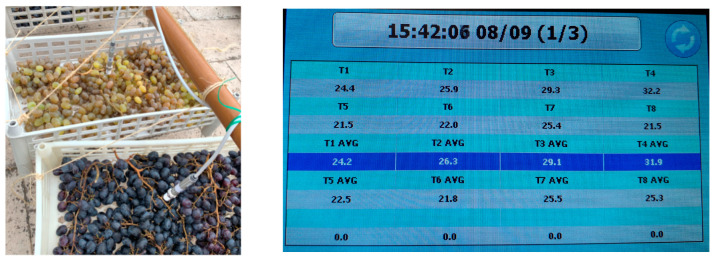
The setup for monitoring the surface temperature of grapes (**left**) and the reading of infrared sensors through an internet address (**right**).

**Figure 2 toxins-15-00613-f002:**
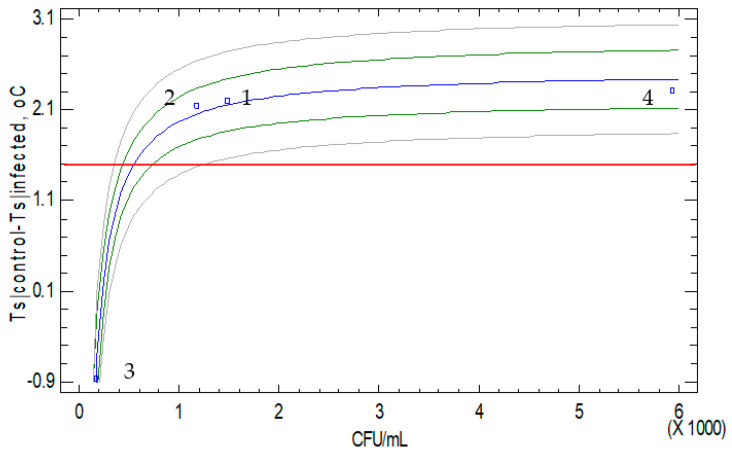
T_s_ difference between control and infected grapes for 24 h monitoring vs. average CFU/mL for all cases (1: open-air drying/white grapes; 2: open-air drying/red grapes; 3: tunnel drying/white grapes; and 4: tunnel drying/red grapes). The red line is the temperature accuracy of the infrared sensors, 1.5 °C. The green and grey lines are the confidence and prediction limits respectively. Blue line is the predicted line based on the equation, T_s_|_control_ − T_s_|_infected_ = 2.532 − 563.241/CFU where $R_{\mathrm{adj}}^{2}$ = 99.40% and SEE = 0118.

**Figure 3 toxins-15-00613-f003:**
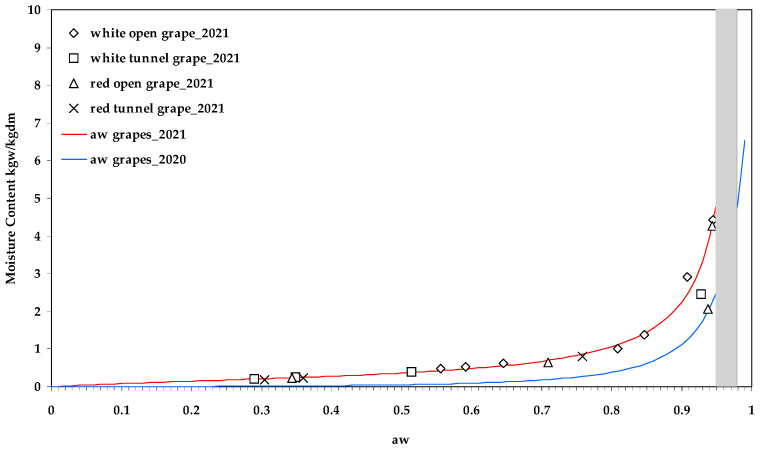
Water activity of white and red drying grapes (open-air and tunnel solar drying) carried out in 2020 (blue line) and 2021 (red line), (points represent experimental data, and lines represent simulated data).

**Figure 4 toxins-15-00613-f004:**
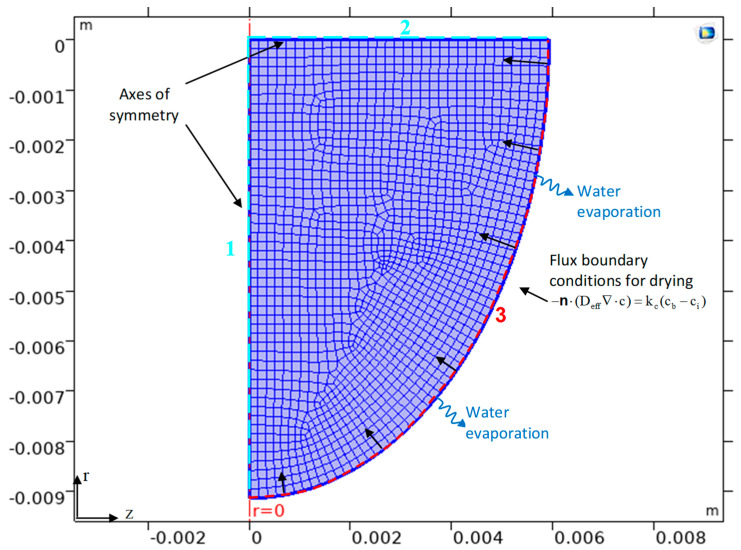
Geometric discretisation of the (shrinking) domain and its boundary conditions, including two axial symmetry conditions (1 and 2) and a moving boundary condition (3). Grey shaded area indicates the optimum area of a_w_ for OTA accumulation.

**Figure 5 toxins-15-00613-f005:**
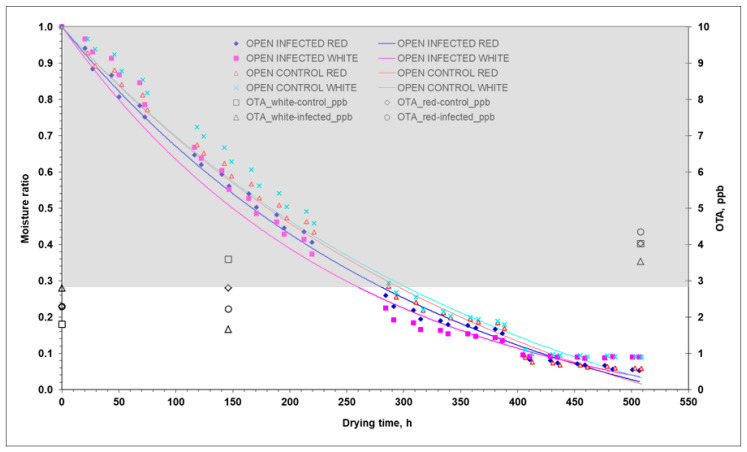
Experimental (points) and predicted (lines) moisture ratios of red and white grapes during open-air drying. The grey highlighted area indicates the optimum zone (MR = 1.0, a_w_ = 0.98; and MR = 0.27, a_w_ = 0.92) for growth of *A. carbonarius,* based on Equation (1). Infected: ${\mathrm{MC}}_{o}^{\mathrm{red}}=$ 7.586 kg_water_/kg_dm_, and ${\mathrm{MC}}_{o}^{\mathrm{white}}=$ 3.766 kg_water_/kg_dm_; Control: ${\mathrm{MC}}_{o}^{\mathrm{red}}=$ 6.093 kg_water_/kg_dm_, and ${\mathrm{MC}}_{o}^{\mathrm{white}}=$ 4.038 kg_water_/kg_dm_.

**Figure 6 toxins-15-00613-f006:**
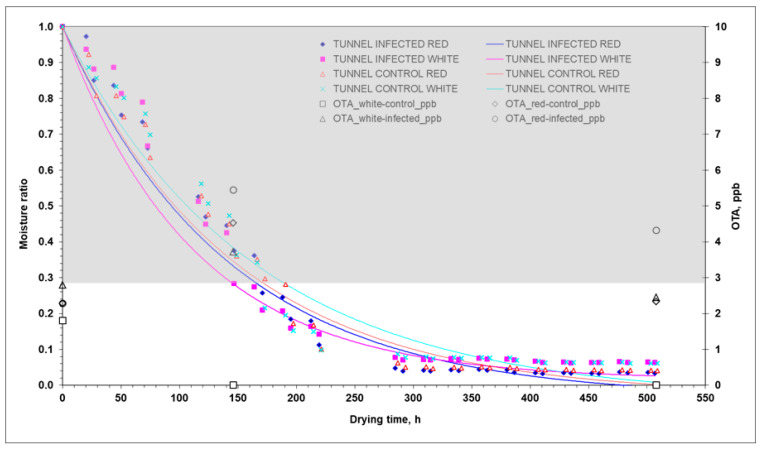
Experimental (points) and predicted (lines) moisture ratios of red and white grapes during tunnel drying. The grey highlighted area indicates the optimum zone (MR = 1.0, a_w_ = 0.98; and MR = 0.27, a_w_ = 0.92) for *A. carbonarius* growth based on Equation (1). Infected: ${\mathrm{MC}}_{o}^{\mathrm{red}}=$ 6.872 kg_water_/kg_dm_, and ${\mathrm{MC}}_{o}^{\mathrm{white}}=$ 3.709 kg_water_/kg_dm_; Control: ${\mathrm{MC}}_{o}^{\mathrm{red}}=$ 5.692 kg_water_/kg_dm_, and ${\mathrm{MC}}_{o}^{\mathrm{white}}=$ 4.001 kg_water_/kg_dm_.

**Table 1 toxins-15-00613-t001:** Average, minimum, and maximum values of the experimental measurements conducted in 2021. Each property was measured for two drying methods (tunnel and open air), two sample treatments (infected and control), and two grape varieties (red—*Crimson* seedless and white—*Thompson* seedless).

Property	Drying Method	Sample Treatment	Grape Variety	Min	Max	Average	Homogeneous Groups						
Grapes surface temperature (°C)	Tunnel	Infected	Red	7.23	65.61	24.63			X	X	X		
			White	3.65	65.40	23.98				X	X		
		Control	Red	6.61	65.07	25.72					X	X	
			White	4.34	62.16	24.76					X	X	
	Open	Infected	Red	6.99	51.66	22.40	X						
			White	8.18	51.02	22.92	X	X					
		Control	Red	9.97	50.73	24.36		X	X				
			White	7.38	55.93	23.34	X	X	X				
Air temperature inside the plastic crates (°C)	Tunnel	Control	Red	4.83	75.24	25.24						X	X
			White	5.32	75.52	25.51							X
	Open	Control	Red	8.18	55.38	24.36			X				
			White	8.12	59.23	24.55			X	X			
Air RH inside the plastic crates (%)	Tunnel		Red	10.62 (5)	93.97 (360)	49.79 (26)					X		
			White	9.025 (6)	87.97 (396)	47.93 (27)			X		X		
	Open		Red	12.24 (7)	94.06 (117)	46.69 (21)			X				
			White	16.76 (7)	91.08 (146)	48.35 (22)			X				
Air RH in distance from the plastic crates (%)				14.29 (13)	70.30 (78)	41.77 (24)	X						
Solar diffusive radiation, W/m^2^				0.0	425	56.78							
Solar total radiation, W/m^2^				0.0	871	210.5							

Note: Values in parentheses are the absolute humidities (g_w_/kg_da_); within each column, the levels containing X form a group within which there are no statistically significant differences.

**Table 2 toxins-15-00613-t002:** Average and standard error (SE) of grape surface temperature during 24 h monitoring during 2021 experiments as difference between control and infected grapes. Total solar radiation for the 2020 and 2021 experiments is also tabulated.

Hour	Open-Air Drying/ White Grapes		Open-Air Drying/ Red Grapes		Tunnel Drying/ White Grapes		Tunnel Drying/ Red Grapes		Average Total Solar Radiation (W/m^2^)	
	Average	SE	Average	SE	Average	SE	Average	SE	2021	2020
1	−0.58	0.16	1.48	0.19	1.74	0.18	0.95	0.23	0.0	0.0
2	−0.72	0.17	1.41	0.17	1.44	0.17	0.66	0.22	0.0	0.0
3	−0.71	0.17	1.37	0.18	1.16	0.15	0.43	0.20	0.0	0.0
4	−0.64	0.18	1.45	0.19	0.98	0.14	0.26	0.19	0.0	0.0
5	−0.77	0.19	1.45	0.19	0.91	0.14	0.14	0.16	0.0	0.0
6	−0.81	0.18	1.41	0.19	0.81	0.12	0.05	0.17	0.0	0.0
7	−0.93	0.19	1.32	0.21	0.67	0.12	−0.03	0.16	21.0	20.64
8	−0.85	0.21	1.27	0.23	0.51	0.13	−0.05	0.14	168.2	174.3
9	0.30	0.17	1.59	0.18	−0.87	0.15	−0.69	0.20	347.4	345.9
10	3.56	0.61	3.17	0.57	−6.13	0.73	−1.86	0.77	538.3	514.7
11	4.31	0.63	4.47	0.67	−8.26	0.84	−0.63	1.03	661.9	659.3
12	3.82	0.58	3.03	0.76	−7.03	0.85	0.40	0.99	696.1	748.7
13	3.10	0.62	1.39	0.54	−4.01	0.69	1.82	0.64	687.9	731.8
14	2.03	0.58	−0.50	0.45	1.36	0.76	3.46	0.65	642.3	684.2
15	0.90	0.53	0.74	0.38	1.98	0.80	4.93	0.44	550.6	596.3
16	2.57	0.46	1.17	0.36	−0.89	0.72	5.77	0.41	414.5	455.0
17	3.26	0.46	1.86	0.34	0.08	0.58	5.14	0.33	270.6	271.8
18	2.97	0.32	3.20	0.34	3.45	0.36	4.50	0.32	90.35	102.8
19	1.68	0.20	3.46	0.37	4.73	0.41	4.37	0.36	2.21	2.96
20	0.99	0.17	3.00	0.37	4.23	0.30	3.01	0.30	0.0	0.0
21	0.26	0.15	2.27	0.32	3.87	0.27	2.22	0.28	0.0	0.0
22	−0.27	0.15	1.78	0.27	3.41	0.25	1.77	0.23	0.0	0.0
23	−0.47	0.16	1.55	0.23	2.70	0.23	1.47	0.23	0.0	0.0
24	−0.56	0.17	1.52	0.21	2.09	0.20	1.14	0.22	0.0	0.0

**Table 3 toxins-15-00613-t003:** Estimates of *G.A.B*. sorption isotherm.

Parameter	Estimate	Asymptotic SE	Asymptotic Confidence 95.0%	
			Lower Interval	Upper Interval
M_o_	0.2176	0.003	0.2106	0.2246
C	4.6244	1.337	1.7360	7.5129
K_b_	1.0060	0.0002	1.0055	1.0065

**Table 4 toxins-15-00613-t004:** Average values of *A. carbonarius* and OTA in grapes sampled at the beginning, middle, and end of the 2021 experiments on red and white grapes, dried in open air and in tunnel. The values in parentheses are from 2020 experiments, where CFU/mL and OTA analyses were performed only on infected samples.

Variety	Infection	Drying	Sampling (h)	CFU (CFU/mL)	OTA (µg/kg)
White	Control	Open	0	133	1.81
			146	700	3.59
			508	5417	4.02
		Tunnel	0	133	1.81
			146	22	0.00
			508	0	0.00
	Infected	Open	0	133 (40)	2.80 (0.00)
			146	1783 (364.4)	1.66 (2.14)
			508	709 (6472.6)	3.54 (4.39)
		Tunnel	0	133 (38.9)	2.80 (0.00)
			146	633 (5.6)	3.73 (4.04)
			508	78 (4.4)	2.45 (12.51)
Red	Control	Open	0	244	2.28
			146	1783	2.80
			508	2673	4.02
		Tunnel	0	244	2.28
			146	17,219	4.52
			508	108	2.33
	Infected	Open	0	244 (264.4)	2.28 (0.00)
			146	300 (14,539.4)	2.21 (6.21)
			508	1823 (8249.6)	4.34 (7.00)
		Tunnel	0	244 (233.9)	2.28 (0.00)
			146	13,875 (112.8)	5.45 (430)
			508	3876 (56.7)	4.32 (8.57)

**Table 5 toxins-15-00613-t005:** Average values of *A. carbonarius* and OTA in grapes sampled at the beginning, middle, and end of the experiments, in red and white grapes, dried in open air and tunnel. Grape variety, drying method, and sampling date were considered as factors in the ANOVA. Data in parentheses are from the 2020 experiments, where CFU/mL and OTA analyses were carried out only on infected samples.

	CFU (CFU/mL)	OTA (µg/kg)
**[1]: Variety**	n.s.	*
White	822.97 (1154.32)	2.34 ^a^ (4.08)
Red	3552.94 (3909.48)	3.26 ^b^ (4.53)
**[2]: Infection**	n.s.	*
Control	2389.84	2.5 ^a^
Infected	1986.07	3.15 ^b^
**[3]: Drying**	n.s.	n.s.
Open-air	1328.67 (4988.43)	2.94 (3.50)
In tunnel	3047.25 (75.37)	2.66 (5.11)
**[4]: Sampling**	n.s.	n.s.
Start	188.92 (144.31)	2.28 (0.58)
Middle	4539.46 (3755.56)	2.99 (4.17)
End	1835.49 (3695.83)	3.13 (8.16)
**Interactions**		
[1] × [2]	n.s.	n.s.
[1] × [3]	*	*
[1] × [4]	n.s	n.s.
[2] × [3]	n.s	*
[2] × [4]	n.s	n.s.
[3] × [4]	n.s	*
[1] × [2] × [3]	n.s	n.s.
[1] × [2] × [4]	n.s	n.s.
[1] × [3] × [4]	n.s	n.s.
[2] × [3] × [4]	n.s	n.s.
[1] × [2] × [3] × [4]	n.s	n.s.

*: *p* ≤ 0.05; n.s. = not significant. Different letters indicate significant differences according to LSD test.

**Table 6 toxins-15-00613-t006:** Shape factor and predicted average of the *Weibull* distribution for *A. carbonarius* and OTA during grape drying experiments in 2021 (of red and white grapes, open-air and tunnel drying). The D-value, which is the modified *Kolmogorov–Smirnov* D-value, utilized as a goodness of fit criterion, was employed.

Variety	Infection	Drying	CFU (CFU/mL)			OTA (µg/kg)		
			Shape Factor	Predicted Average	D-Value	Shape Factor	Predicted Average	D-Value
White	Control	Open	0.531	1094.41	0.225	3.662	3.522	0.216
		Tunnel	2.619	117.50	0.385	10.752	2.835	0.385
	Infected	Open	0.657	1063.40	0.225	4.227	3.411	0.216
		Tunnel	1.056	281.91	0.385	5.554	2.991	0.385
Red	Control	Open	0.779	2031.18	0.301	5.189	3.899	0.262
		Tunnel	0.382	4938.97	0.236	2.625	3.435	0.295
	Infected	Open	1.275	981.41	0.301	8.376	3.800	0.262
		Tunnel	0.704	6787.15	0.236	5.489	4.524	0.295

**Table 7 toxins-15-00613-t007:** Shrinkage velocities of the minor x and major y axes of the grapes.

Drying	Infection	Variety	Shrinkage Velocity × 10^−9^ (m/s)	
			*x*-Axis	*y*-Axis
Open-air	Control	White	2.449	1.669
		Red	2.090	1.483
	Infected	White	3.165	2.625
		Red	2.888	2.051
Tunnel	Control	White	4.009	3.235
		Red	2.476	1.576
	Infected	White	4.007	2.431
		Red	3.859	1.704

**Table 8 toxins-15-00613-t008:** Estimated k_c_, MRE, D_eff_, and r_peel_ for the drying cases.

Drying	Infection	Variety	k_c_ × 10^−9^ (m/s)	r_peel_ × 10^8^ (s/m)	MRE (%)	$D_{\mathrm{eff}}$× 10^−11^ (m^2^/s)
Open-air	Control	White	5.05 (2.75)	1.98 (3.64)	3.42 (2.17)	0.70 (14.8)
		Red	2.94 (3.33)	3.40 (3.00)	4.65 (3.64)	1.80 (29.9)
	Infected	White	4.18 (3.16)	2.39 (3.16)	3.05 (2.55)	3.60 (50.3)
		Red	3.31 (2.61)	3.02 (3.83)	3.36 (5.53)	2.10 (21.5)
Tunnel	Control	White	5.18 (6.14)	1.93 (1.63)	3.88 (6.20)	4.50 (10.6)
		Red	5.17 (6.37)	1.93 (1.57)	5.72 (6.74)	3.40 (39.5)
	Infected	White	5.85 (8.31)	1.71 (1.20)	4.38 (6.72)	8.60 (53.8)
		Red	5.17 (7.79)	1.93 (1.28)	4.15 (8.30)	3.50 (151)

## Data Availability

Not applicable.

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
