# Peer review of "IoT for Monitoring Fungal Growth and *Ochratoxin A* Development in Grapes Solar Drying in Tunnel and in Open Air"

_toxins, 2023, doi:10.3390/toxins15100613_

Round 1

Reviewer 1 Report

Overall, the novel part of this manuscript is using Internet of Things (IoT) technology and infrared sensors to monitor the solar drying process for reducing fungal growth and Ochratoxin A (OTA) contamination of raisins. However, the manuscript cannot be published without revision due to following issues.

Abstract: Please provide a brief conclusion regarding the feasibility of IoT and infrared sensor to monitor solar drying of grapes based on your results. Which drying method do you recommend, the open air drying and tunnel drying?

Introduction:

1. The introduction should justify why open air drying and tunnel drying were used in this study.

2. Page 1: change driven to caused

3. Page 2, first paragraph: change “traced” to “detected”. change " The region of origin of the grapes" to "The grape growing region"

4. What does "positive gradient" mean?

5. Please clarify if A. carbonarius is black Aspergilli or not? What is “this knowledge”?

6. Mehri et al. [4}...(Raisins, wines etc). This is basic the same as " Factors significantly affecting fungal growth ...." and I suggest delete it.

7. the interrelationship mechanisms ? What does that mean? Please clarify.

Results and Discussion:

1.      Results part is difficult to understand. The Results were not clearly presented and the description of results is confusing.

2.       Table 1: (1) The title of the table is unclear. what does the "values" mean? The Table and figure titles should be self-explainable without referring to the text. (2) Data in Table 1 are not well presented and are difficult to figure out which data correspond to what treatment.

3.       Page 4: Authors stated that “This experimental setup introduces a bias in the surface measurement, as uncontrolled environmental conditions can affect the accuracy of the sensors e.g. short-standing air currents cooling the surface of the grapes”. This is a limitation of the study and should be mentioned in the conclusion and possible improvement to eliminate the bias should be proposed.

4.       Page 5: what do "The microbial load (CFU/mL) per case" and "six CFU/mL values per case" mean ?

5.       Table 3: C and K are adsorption constants, but what is the difference between them?

6.       able 4: It would be more informative if the water activities and temperature samples at each sampling time were provided in this table as the water activity changes with temperature. Aw and air temperature also have detrimental impacts on mold growth and OTA production.

7.       Page 8: what do you mean "the CFU/mL and OTA values came from a Weibull distribution"? Do you mean the CFU/mL and OTA contents calculated using Weibull distribution?

8.       Page 9: Shape of factor:: How the 5.7 times and 7.0 times were calculated?

Conclusion

The conclusion is not well written.  It read more like the summary of results. The conclusion should emphasize the applicability of IoT for monitoring mold growth and OTA during open air drying and tunnel drying of grapes. Its advantages and the need for improvement, etc based on the results of this study. Without this, the manuscript will be similar to what is published in the Toxins in 2021 (https://doi.org/10.3390%2Ftoxins13060400). The limitation of this study and future research should also be included in the conclusion.

In conclusions, line 3: change "he" to "The"

Materials and Methods

1.       Please clarify if the grapes for drying were placed as monolayer or multi-layers. If multilayers, the temperature and moisture contents on the surface, in middle and at bottom could be significantly different. The surface temperature cannot represent mid and bottom temperature.

2.       Page 15: change "After grape sampling, the samples per tested case" to "After grape sampling, the grapes from different treatments  or groups were stored separately in plastic bags at -18C".

3.       should the "elsewhere" in equation 2-Weibull Distribution  be  X 0 ? Make it aligned with X>0.

Author Response

Comments and Suggestions for Authors (Corrections in red)

Overall, the novel part of this manuscript is using Internet of Things (IoT) technology and infrared sensors to monitor the solar drying process for reducing fungal growth and Ochratoxin A (OTA) contamination of raisins. However, the manuscript cannot be published without revision due to following issues.

Abstract: Please provide a brief conclusion regarding the feasibility of IoT and infrared sensor to monitor solar drying of grapes based on your results. Which drying method do you recommend, the open air drying and tunnel drying? Further discussion has been added at the end of the abstract.

Introduction:

  1. The introduction should justify why open air drying and tunnel drying were used in this study. Discussion has been added.
  2. Page 1: change driven to caused. Corrected.
  3. Page 2, first paragraph: change “traced” to “detected”. change " The region of origin of the grapes" to "The grape growing region" Corrected
  4. What does "positive gradient" mean? Phrase was corrected
  5. Please clarify if A. carbonarius is black Aspergilli or not? What is “this knowledge”? Explanations have been added.
  6. Mehri et al. [4}...(Raisins, wines etc). This is basic the same as " Factors significantly affecting fungal growth ...." and I suggest delete it. This paragraph was erased.
  7. the interrelationship mechanisms ? What does that mean? Please clarify. The sentence was corrected to “underlined mechanisms”.

Results and Discussion:

  1.     Results part is difficult to understand. The Results were not clearly presented and the description of results is confusing. Corrections have been done to improve meaning.
  2. Table 1: (1) The title of the table is unclear. what does the "values" mean? The Table and figure titles should be self-explainable without referring to the text. (2) Data in Table 1 are not well presented and are difficult to figure out which data correspond to what treatment. Corrections have been made in Table 1.
  3. Page 4: Authors stated that “This experimental setup introduces a bias in the surface measurement, as uncontrolled environmental conditions can affect the accuracy of the sensors e.g. short-standing air currents cooling the surface of the grapes”. This is a limitation of the study and should be mentioned in the conclusion and possible improvement to eliminate the bias should be proposed. This phrase was moved to “Conclusion” section and further discussion was added.
  4. Page 5: what do "The microbial load (CFU/mL) per case" and "six CFU/mL values per case" mean ? The phrase was corrected to improve meaning.
  5. Table 3: C and K are adsorption constants, but what is the difference between them? Explanation was added.
  6. able 4: It would be more informative if the water activities and temperature samples at each sampling time were provided in this table as the water activity changes with temperature. Aw and air temperature also have detrimental impacts on mould growth and OTA production. We agree, although due to the high number of samples (489 berries per sampling time) the temperature and water activity measurements per berry are not feasible at the sampling time. Instead the OTA values, where placed together with the drying curves in Figures 5 and 6 while in Figure 3 aw and moisture content is presented.
  7. Page 8: what do you mean "the CFU/mL and OTA values came from a Weibull distribution"? Do you mean the CFU/mL and OTA contents calculated using Weibull distribution? The phrase corrected.
  8. Page 9: Shape of factor:: How the 5.7 times and 7.0 times were calculated? Explanation and rephrasing was added.

Conclusion

The conclusion is not well written.  It read more like the summary of results. The conclusion should emphasize the applicability of IoT for monitoring mold growth and OTA during open air drying and tunnel drying of grapes. Its advantages and the need for improvement, etc based on the results of this study. Without this, the manuscript will be similar to what is published in the Toxins in 2021 (https://doi.org/10.3390%2Ftoxins13060400). The limitation of this study and future research should also be included in the conclusion. The “Conclusions” section was rewritten in the light of the reviewer’s suggestion.

In conclusions, line 3: change "he" to "The" Correction was applied.

Materials and Methods

  1. Please clarify if the grapes for drying were placed as monolayer or multi-layers. If multilayers, the temperature and moisture contents on the surface, in middle and at bottom could be significantly different. The surface temperature cannot represent mid and bottom temperature. Explanation was added.
  2. Page 15: change "After grape sampling, the samples per tested case" to "After grape sampling, the grapes from different treatments or groups were stored separately in plastic bags at -18C". Correction was added
  3. should the "elsewhere" in equation 2-Weibull Distribution be  X ≤ 0 ? Make it aligned with X>0. Correction was added.

Reviewer 2 Report

This work comparing quality changes (infection with aspergillus and production of toxins and ochratoxin) of batches of white grapes or black grapes in two drying processes for the production of raisins, is clearly described, however there are some inaccuracies and errors in the conclusion particularly in the first paragraph.

the tests were only developed in 2 consecutive years, which may be directly related to the lack of significantly different results in several parameters. There should be more data from more years so that the results can be more robust.

The bibliography in general is not very recent, since the most recent reference is a legislative regulation and the remaining 5 most recent references are 2 or more years old, with the majority of other references being 10 to 15 years old.

Author Response

Comments and Suggestions for Authors (Corrections in blue)

This work comparing quality changes (infection with aspergillus and production of toxins and ochratoxin) of batches of white grapes or black grapes in two drying processes for the production of raisins, is clearly described, however there are some inaccuracies and errors in the conclusion particularly in the first paragraph.

The tests were only developed in 2 consecutive years, which may be directly related to the lack of significantly different results in several parameters. There should be more data from more years so that the results can be more robust. The second-year experiments assessed the infrared sensing and IoT protocol used to monitor grape surface temperature to detect fungal growth early. The lack of significance in temperature sensing is subject of the infrared sensing protocol and this is discussed in “Conclusions” section.

The bibliography in general is not very recent, since the most recent reference is a legislative regulation and the remaining 5 most recent references are 2 or more years old, with the majority of other references being 10 to 15 years old. Selective changes have been applied in bibliography. The oldest ones are from well-known handbooks that are used in food modelling (stochastic and deterministic). The bibliography regarding such complex subject regarding grape drying and fungus growth is limited.

Reviewer 3 Report

The authors reported an analysis of various aspects of solar drying for grapes dried in open-air and in tunnels in this work. It is of great significance for the study to collect the data. The results will provide essential information on the mechanisms that promote the drying process, A. carbonarius development, and OTA accumulation.

However, there were some errors in writing and questions in the manuscript that the authors should explain before this work can be accepted.

Page 2 paragraph 2, “The region of origin of the grapes”, the expression should be corrected.

Page 14 paragraph 2, “he grapes’ surface temperature”, the expression should be corrected to “The grapes’ surface temperature”.

References in this article should be updated, such as within the last two or three years. This helps provide evidence for experiments.

Author Response

Comments and Suggestions for Authors (Corrections in bold)

"The authors reported an analysis of various aspects of solar drying for grapes dried in open-air and in tunnels in this work. It is of great significance for the study to collect the data. The results will provide essential information on the mechanisms that promote the drying process, A. carbonarius development, and OTA accumulation.

However, there were some errors in writing and questions in the manuscript that the authors should explain before this work can be accepted.

  • Page 2 paragraph 2, “The region of origin of the grapes”, the expression should be corrected. Corrected according to reviewers 1 suggestion to “The grape growing region”

  • Page 14 paragraph 2, “he grapes’ surface temperature”, the expression should be corrected to “The grapes’ surface temperature”. Corrected

  • References in this article should be updated, such as within the last two or three years. This helps provide evidence for experiments." References have been updated.

Round 2

Reviewer 1 Report

The readability of revised manuscript 2618784-R1 is improved, but more grammar errors and other issues are identified during review of revised manuscript. Please make corrections and address the issues accordingly.

Page 2, the last paragraph of introduction. Since the manuscript described the completed research work, the verbs should be in paste tense, not future tense. Please check and correct.

Page 8: I think “CFU/mL and OTA values are derived from a Weibull distribution"  should be "The  distribution of CFU/mL and OTA values during drying fall into Weibull distribution" 

Page 12: "Figures 5 and 6 show the drying curves [MR=f(t)] with the OTA" . This sentence is unclear, please rephrase it.

Fig.6. The changes of moisture data with drying time did not fit the curve well. Maybe the moisture data of grapes in  tunnel drying do not follow the same pattern as those in open air drying.   Please consider possible better predictive model.

Page 14, Conclusion:

1.       change  "Achieving reliable and reproducible grape surface temperatures through the use of infrared sensing requires adherence to ..." to " To achieve reliable and reproducible grape surface temperatures through the use of infrared sensing,  It requires adherence to..."

2.       Conclusion: "From this correlation was found that once the CFU/mL values fall below a certain threshold, the resulting difference in grape surface temperature (Ts|control-Ts|infected) is equal to or less than the accuracy of the infrared sensors”. Unclear sentence, please rephrase it.

3.       What is CFD simulation? It was not mention in Materials and Methods section, and other sections before this point.

4.       Should "feeding" be "fitting"?

5.       Estimating the Deff and rpeel aided in understanding the mechanisms that promote A. carbonarius growth and OTA production, possibly attributing this to water transportion from the endocarp and mesocarp of the grapes to their skin (exocarp) which initiates A. carbonarius proliferation. Confusing sentence, please rewrite it.

A proof reading is recommended.

Author Response

Comments and Suggestions for Authors (Corrections in red and underlined)

The readability of revised manuscript 2618784-R1 is improved, but more grammar errors and other issues are identified during review of revised manuscript. Please make corrections and address the issues accordingly.

Page 2, the last paragraph of introduction. Since the manuscript described the completed research work, the verbs should be in paste tense, not future tense. Please check and correct. Corrections were done in the last paragraph of “Introduction”

Page 8: I think “CFU/mL and OTA values are derived from a Weibull distribution" should be "The  distribution of CFU/mL and OTA values during drying fall into Weibull distribution"  Corrected

Page 12: "Figures 5 and 6 show the drying curves [MR=f(t)] with the OTA" . This sentence is unclear, please rephrase it. The sentence was rephrased.

Fig.6. The changes of moisture data with drying time did not fit the curve well. Maybe the moisture data of grapes in tunnel drying do not follow the same pattern as those in open-air drying. Please consider possible better predictive model. The dispersion of the experimental moisture content around the predicted line (drying curve) was mainly found in the “Tunnel Control White” case. Field experiments are subjected in uncontrollable drying conditions and thus no deterministic model can simulate accurately the derived drying curves.

Page 14, Conclusion:

  1. Change "Achieving reliable and reproducible grape surface temperatures through the use of infrared sensing requires adherence to ..." to " To achieve reliable and reproducible grape surface temperatures through the use of infrared sensing, It requires adherence to..." Corrected
  2. Conclusion: "From this correlation was found that once the CFU/mL values fall below a certain threshold, the resulting difference in grape surface temperature (Ts|control-Ts|infected) is equal to or less than the accuracy of the infrared sensors”. Unclear sentence, please rephrase it. The sentence was rephrased.
  3. What is CFD simulation? It was not mention in Materials and Methods section, and other sections before this point. Corrected to Computational Fluid Dynamics (CFD).
  4. Should "feeding" be "fitting"? Corrected to “feeding as inputs”.
  5. Estimating the Deff and rpeel aided in understanding the mechanisms that promote A. carbonarius growth and OTA production, possibly attributing this to water transportion from the endocarp and mesocarp of the grapes to their skin (exocarp) which initiates A. carbonarius proliferation. Confusing sentence, please rewrite it. The sentence was rewritten to “Estimating Deff and rpeel can help in comprehending the mechanisms involved in growth of A. carbonarius and OTA production. These two properties describe the movement of water from the endocarp and mesocarp of grapes to their skin (exocarp)”.

Reviewer 3 Report

Accept in present form.

Author Response

Thank you
